# Federated Circuits: A Unified Framework for Scalable and Efficient Federated Learning

**Jonas Seng**[1,*]**, Florian Peter Busch**[1,2]**, Pooja Prasad**[3]
**Devendra Singh Dhami**[3]**, Martin Mundt**[5] **Kristian Kersting**[1,2,4]

[1]Department of Computer Science, Technical University of Darmstadt, Germany
[2]Hessian Center for AI (hessian.AI), Germany
[3]Dept. of Mathematics and Computer Science, Eindhoven University of Technology, Netherlands
[4]German Research Center for AI (DFKI), Germany
[5] Department of Mathematics and Computer Science, University of Bremen
* `jonas.seng@tu-darmstadt.de`

## Abstract

Probabilistic circuits (PCs) enable us to represent joint distributions over a set of random variables and can be seen as hierarchical mixture models. This representation allows for various probabilistic queries to be answered in tractable time. However, the properties of PCs so far have only been explored in the realm of tractable probabilistic modeling. In this work, we unveil a deep connection between PCs and federated learning (FL), leading to federated circuits (FCs)—a novel, flexible, modular, and communication-efficient federated learning (FL) framework that unifies for the first time horizontal, vertical, and hybrid FL in one framework by re-framing FL as a density estimation problem over distributed datasets. Also, FCs allow us to scale *tractable* probabilistic models (PCs) to large-scale datasets by recursively partitioning datasets and the model itself across a distributed learning environment. We empirically demonstrate FC's versatility in handling horizontal, vertical, and hybrid FL within a unified framework on multiple classification tasks. Further, we demonstrate FCs' capabilities to scale PCs to large-scale datasets on various real-world image datasets.

## 1 Introduction

Probabilistic Circuits (PCs) are a family of models that provide tractable inference for various probabilistic queries (Poon & Domingos, 2011; Choi et al., 2020). This is achieved by representing a joint distribution by a computation graph on which certain structural properties are imposed. These properties of PCs, as well as further optimizations like improving the compactness of PC representations and tailoring them to specific hardware architectures (Peharz et al., 2020a; Liu et al., 2024), provide significant computational advantages over traditional probabilistic models such as Bayesian networks (Pearl, 1985) and allow tractable probabilistic modeling of larger datasets. However, the model properties of PCs not only allow tractable probabilistic modeling but also have a deep connection to federated learning (FL). For popular models like neural networks and tree-based models, designing architectures and training schemes specifically tailored to horizontal, vertical, and hybrid FL is necessary. This makes existing FL frameworks inflexible and only applicable to certain scenarios. Designing a framework that unifies all three FL scenarios is considered a non-trivial endeavor in the Fl community Li et al. (2023a); Wen et al. (2023). Inspired by probabilistic circuits (PCs), we propose *federated circuits (FCs)*, a novel, flexible, modular, and communication-efficient FL framework that is capable of handling horizontal, vertical, and hybrid FL scenarios at the same time. This is achieved by considering FL as a density estimation task and designing a distributed and modular model architecture akin to PCs: In PCs, sum nodes combine probability distributions over the same set of variables via a mixture. This resembles the horizontal FL setting (Konečný et al., 2016; Li et al., 2020), where all clients hold the same features but different samples. In contrast, the case of vertical FL (Yang et al., 2019; Wu et al., 2020) in which the same samples are shared, but features are split across clients, can be linked to the product nodes used in PCs, which combine

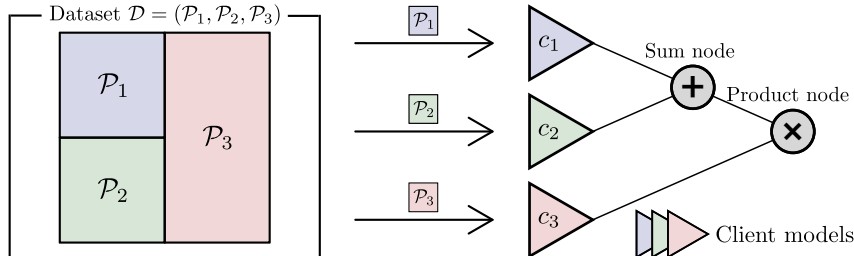

Figure 1: **Federated Circuits unify horizontal, vertical, and hybrid FL.** FCs reframe FL as a density estimation task and learn a joint model over a dataset $\mathcal{D}$ split into a set of $n$ partitions $\{\mathcal{P}_i\}_{i=1}^n$ s.t. $\mathcal{D} = \bigcup_{i=1}^n \mathcal{P}_i$. Each partition is held by a client (i.e., machine) $c_j$, and the resulting federated circuit (FC) is learned jointly. This approach also allows to scale up tractable probabilistic models.

distributions of a disjoint set of variables. Consequently, the hybrid FL (Zhang et al., 2020) setting, where both samples and features are separated across clients, can be represented by a combination of sum and product nodes.

FCs naturally handle all three FL settings and, therefore, also provide a flexible way of scaling up PCs by learning a joint distribution over a dataset arbitrarily partitioned across a set of clients (see Fig. 1 for an illustration). Imposing the same structural properties as for PCs, FCs achieve tractable computation of probabilistic queries like marginalization and conditioning across multiple machines. To learn FCs, we propose a highly communication-efficient learning algorithm that leverages the semi-ring structure within the design of FCs. Our experimental evaluation shows that FCs outperform EiNets (Peharz et al., 2020a) and PyJuice (Liu et al., 2024) on large-scale density estimation tasks, demonstrating the benefits of scaling up PCs. Additionally, FCs outperform or achieve competing results on various classification tasks in all federated settings compared to state-of-the-art neural network-based and tree-based methods, demonstrating its effectiveness and flexibility in FL. We make the following contributions: **(1)** We introduce FCs, a modular, communication-efficient, and scalable FL framework unifying horizontal, vertical, and hybrid FL by mapping the semantics of PCs to FL. **(2)** We practically instantiate FCs to FedPCs and demonstrate how the FC framework can be leveraged to scale up PCs to large real-world datasets. **(3)** We propose a one-pass training scheme for FedPCs that is compatible with a broad set of learning algorithms. **(4)** We provide extensive experiments demonstrating the effectiveness of our approach for learning large-scale PCs and performing FL. We consider classification and density estimation on tabular and image data. Our code is publicly available at `https://github.com/J0nasSeng/federated-spn`.

## 2 PRELIMINARIES AND RELATED WORK

In the following, we briefly introduce PCs and FL and give an overview of relevant related work.

**Probabilistic Circuits (PCs).** PCs encode a probability distribution as a computation graph that allows for tractable inference of a wide range of queries such as conditioning and marginalization. The computation graph of PCs consists of sum nodes computing a mixture distribution over a set of random variables (RVs) and product nodes computing product distributions over sets of RVs (see App. A.1 for details). Several works have tackled the goal of scaling PCs. Peharz et al. (2020b) have shown that learned PC structures can be replaced by large, random structures to scale to larger problems. Changes in the model layout, such as parallelizable layers via einsum-operations (Peharz et al., 2020a) and a reduction in IO operations (Liu et al., 2024), were also shown to reduce the speed of computation drastically. Liu et al. (2022) improved the performance of PCs by latent variable distillation using deep generative models for additional supervision during learning.

**Federated Learning (FL).** The goal of FL is to collaboratively learn an ML model across clients without sharing the clients' data. One distinguishes between horizontal, vertical, and hybrid FL depending on how data is partitioned. In horizontal FL, a dataset $\mathbf{D} \in \mathbb{R}^{n \times d}$ is partitioned s.t. each client holds the same $d$ features but different, non-overlapping sets of samples. In vertical FL, $\mathbf{D}$ each client holds the same $n$ samples but different, non-overlapping subsets of the $d$ features. Hybrid FL combines the characteristics of horizontal and vertical FL (Wen et al., 2023; Li et al., 2023a). For all three FL settings, specifically tailored methods have been proposed to enable collaborative

learning of models. The most common scheme in horizontal FL is to average the models of all clients regularly during training (McMahan et al., 2016; Karimireddy et al., 2020a;b; Sahu et al., 2018), while in vertical and hybrid FL, tree-based and neural models are the predominant choice and are typically learned by sharing data statistics or feature representations among clients (Kourtellis et al., 2016; Cheng et al., 2021; Vepakomma et al., 2018; Liu et al., 2019; Ceballos et al., 2020; Chen et al., 2020; Li et al., 2023b; 2024). Especially in hybrid FL, however, complex training procedures are required that are highly specialized towards certain model classes.

## 3 FEDERATED CIRCUITS

In the following, we present federated circuits (FCs), an elegant framework unifying horizontal, vertical, and hybrid FL. Also, we show how FCs can be leveraged to scale up PCs effectively.

**Problem Statement & Modeling Assumptions.** Given a dataset $\mathbf{D}$ and a set of clients $\mathcal{C}$ where each $c \in \mathcal{C}$ holds a partition $\mathbf{D}_c$ of $\mathbf{D}$; we aim to learn the joint distribution $p(\mathbf{X})$ over random variables $\mathbf{X}$ (i.e., the features of $\mathbf{D}$). The partitioning of $\mathbf{D}$ is not further specified. Hence, each client might only hold a subset of random variables $\mathbf{X}_c \subseteq \mathbf{X}$ with support $\mathcal{X}_c$. This can be interpreted as each $c \in \mathcal{C}$ holding a dataset $\mathbf{D}_c \sim p_c$ where $p_c$ is a joint distribution over $\mathbf{X}_c$ which is related to $p(\mathbf{X})$. We introduce two critical modeling assumptions relevant for learning a joint distribution $p(\mathbf{X})$ from a dataset $\mathbf{D}$ partitioned across a set of machines.

**Assumption 1** (Mixture Marginals). *There exists a joint distribution $p$ such that the relation $\int_{\mathbf{X} \setminus \mathbf{X}_S} p(x) = \sum_{l \in L} q(L = l) \cdot p_S(x|L = l)$ holds for all $x \in \mathcal{X}$. Here, $\mathbf{X}_S \subseteq \mathbf{X}$ is a subset of the union of client random variables $\mathbf{X} = \cup_{c \in \mathcal{C}} \mathbf{X}_c$. Further, $\mathcal{X} = \bigtimes_{c \in \mathcal{C}} \mathcal{X}_c$ is the support of $\mathbf{X}$, each $p_S$ is defined over $\mathbf{X}_S \subseteq \mathbf{X}$ and $q$ is a prior over a latent $L$.*

**Assumption 2** (Cluster Independence). *Given disjoint sets of random variables $\mathbf{X}_1, \cdots, \mathbf{X}_n$ and a joint distribution $p(\mathbf{X}_1, \cdots, \mathbf{X}_n)$, assume that a latent $L$ can be introduced s.t. the joint can be represented as $p(\mathbf{X}_1, \cdots, \mathbf{X}_n) = \sum_l q(L = l) \prod_{i=1}^n p(\mathbf{X}_i|L = l)$ where $q$ is a prior distribution over the latent $L$.*

Assumptions 1 and 2 ensure that the client's data is sufficient to learn the joint $p(\mathbf{X})$ and that dependencies among features residing on different clients can be learned. See App. B for a discussion.

### 3.1 BRIDGING PROBABILISTIC CIRCUITS AND FEDERATED LEARNING

We now illustrate an inherent connection between PC semantics and FL. This will allow us to train PCs on data partitioned over a set of clients and thus greatly increase the scaling potential of PCs.

**Sum Nodes & Horizontal FL.** In horizontal FL, each client is assumed to hold the same set of features, i.e., $\mathbf{X}_c = \mathbf{X}_{c'}$ for all $c, c' \in \mathcal{C}$. However, each client holds different samples. The horizontal FL setting precisely corresponds to the interpretation of sum nodes in PCs: A sum node splits a dataset into multiple disjoint clusters. This results in a mixture distribution representing the data that is learned from the disjoint clusters.

**Definition 1** (Horizontal FL). *Assume a set of samples $\mathbf{D}_c \sim p_c$ on each client $c \in \mathcal{C}$, a joint distribution $p$ adhering to Assumption 1 and that $\mathbf{X}_c = \mathbf{X}_{c'}$ for all $c, c' \in \mathcal{C}$ s.t. $c \neq c'$. We define horizontal FL as fitting a mixture distribution $\hat{p} = \sum_{c \in \mathcal{C}} q(c) \cdot \hat{p}_c$ such that $d(\hat{p}, p)$ and $d(p_c, \hat{p}_c)$ are minimal for all $c \in \mathcal{C}$ where $d$ is a distance metric and $\hat{p}_c$ local distribution estimates.*

In contrast to prominent horizontal FL methods which regularly aggregate *model parameters* during training, we aggregate *distributions*. This modular approach aggregating distributions into mixtures, naturally handles heterogeneous client distributions, leading to robustness against differing client distributions. Also, since clients can train models independently, communication costs are minimized.

**Product Nodes & Vertical FL.** In vertical FL, each client is assumed to hold a disjoint set of features, i.e., $\mathbf{X}_c \cap \mathbf{X}_{c'} = \emptyset$ for all $c, c' \in \mathcal{C}$. In contrast to horizontal FL, all clients hold different features belonging to the same sample instances. There is a semantic connection between vertical FL and PCs. Product nodes in PCs compute a product distribution defined on a disjoint set of random variables, thereby separating the data along the feature dimension. This corresponds to the vertical FL setting. However, a product node assumes the random variables of the child distributions to be independent of

each other. This is an unrealistic assumption for vertical FL, where features held by different clients might be statistically dependent. In Sec. 3.2, we exploit Assumption 2 to capture such dependencies.

**Definition 2** (Vertical FL). *Assume a set of samples $\mathbf{D}_c \sim p_c$ on each data owner $c \in \mathcal{C}$, the existence of a joint distribution $p$ adhering to Assumptions 1 and 2 and that $\mathbf{X}_c \cap \mathbf{X}_{c'} = \emptyset$ holds for all $c, c' \in \mathcal{C}$ s.t. $c \neq c'$. We define vertical FL as estimating a joint distribution $\hat{p}$ s.t. $d(p, \hat{p})$ is minimal and $\int_{\mathbf{X} \setminus \mathbf{X}_c} \hat{p}(x) = \hat{p}_c(x)$ for all $x \in \mathcal{X}$ where $d$ is a distance metric and $\hat{p}_c$ are estimates of client distributions.*

**PCs & Hybrid FL.** Given Defs. 1 and 2, hybrid FL is a combination of both. In terms of PC semantics, this amounts to building a hierarchy of fusing marginals and learning mixtures. Provided with these probabilistic semantics, we can now formally bridge PCs and FL. In the following, we distinguish between clients $\mathcal{C}$ and servers $\mathcal{S}$ and define the set of machines participating in training as $\mathcal{N} = \mathcal{C} \cup \mathcal{S}$. Bringing everything together and abstracting from the probabilistic interpretation, we define **federated circuits** (FCs) as follows.

**Definition 3** (Federated Circuits). *A **federated circuit** (FC) is a tuple $(\mathcal{G}, \psi_{\mathcal{G}}, \omega)$ where $\mathcal{G} = (V, E)$ is a rooted, Directed Acyclic Graph (DAG), $\psi_{\mathcal{G}} : V \to \mathcal{N}$ assigns each $\mathsf{N} \in V$ to a client/server $n \in \mathcal{N}$ based on the structure of $\mathcal{G}$ and $\omega : V \to O$ assigns an operation $o \in O$ to each node $\mathsf{N} \in V$ where $o : dom(\mathrm{ch}(\mathsf{N})) \to dom(\mathsf{N})$ computes the value of $\mathsf{N}$ given the values of the children of $\mathsf{N}$.*

FCs extend the definition of PCs in the sense that FCs represent a computational graph $\mathcal{G} = (V, E)$ distributed over multiple machines where arbitrary operations can be performed in each node $\mathsf{N} \in V$. Note that through $\mathcal{G}$, FCs also define the structure of a communication network among participating machines. Also, FCs are not restricted to a probabilistic interpretation. For example, parameterizing leaves by decision trees yields a bagging model.

## 3.2 Federated Probabilistic Circuits

We now introduce federated PCs (FedPCs), thereby following the probabilistic interpretation from Sec. 3.1. We align the PC structure with the communication network structure to form a FedPC.

**Definition 4** (Federated PC). *A Federated PC (FedPC) is a FC where each leaf node $\mathsf{C}$ is a density estimator and each node $\mathsf{N}$ s.t. $\mathrm{ch}(\mathsf{N}) \neq \emptyset$ is either a sum node ($\mathsf{S}$) or a product node ($\mathsf{P}$).*

Note that only the client nodes $\mathsf{C}$ hold a dataset, and we only demand the clients to be parameterized by a density estimator (e.g., PCs). The assignment function $\phi$ establishes a direct correspondence between PC semantics and the communication network by transforming the PC's computation graph into a distributed computation graph. Hence, $\mathcal{G}$ defines the model's computation and the communication among participants. Inference is performed as usual in PCs by propagating likelihood values from the leaf nodes to the root node. Training FedPCs requires adapting the regular training procedure for PCs because, in FL, clients cannot access other clients' data while popular training algorithms like Expectation Maximization (EM) or LearnSPN Gens & Domingos (2013) assume access to all features and samples. We tackle this with a *one-pass* training procedure for FedPCs.

**One-Pass Training.** Our one-pass learning algorithm learns the structure and parameters of FedPCs such that local models can be trained independently (Algo. 1, Fig. 4). Before training, all clients $c \in \mathcal{C}$ share their set of uniquely identifiable features/random variables $\mathbf{X}_c$ with a server, resulting in the feature set indicator matrix $\mathbf{M}^{|\mathcal{C}| \times |\mathbf{X}|}$ (**Lines 1-2**). Feature identifiers can be names of features such as "account balance" and must correspond to the same random variable on all clients (thus uniquely identifiable). Then, the server divides the joint feature space $\mathbf{X}$ into disjoint subspaces $\mathbf{S}^{(j)}$. For this, we consider the set of distinct column vectors $\mathcal{U}$ of $\mathbf{M}$ where we denote distinct vectors as $\mathbf{u}$. Since each column of $\mathbf{M}$ indicates the set of clients a feature resides on, we can use each $\mathbf{u} \in \mathcal{U}$ to compute a set of features that are shared across the same set of clients. This results in $|\mathcal{U}|$ distinct feature sets, denoted by $\{\mathbf{S}^{(1)}, \dots, \mathbf{S}^{(|\mathcal{U}|)}\}$. Each $O_{\mathbf{S}^{(j)}}$ denotes the set of clients that hold the features in $\mathbf{S}^{(j)}$. (**Lines 3-7**). This procedure is illustrated in Fig. 4(top) in App. C.

Afterward, the FedPC structure is constructed as shown in Fig. 4(bottom) in App. C: First, we build a mixture (sum node) for each subspace $\mathbf{S}^{(j)}$ where $|O_{\mathbf{S}^{(j)}}| > 1$, i.e., more than one client holds $\mathbf{S}^{(j)}$ (**Lines 9-12**). Note that this can be seen as aggregating client modules/models into one large mixture. enabling each client to learn a PC over $\mathbf{S}^{(j)}$ independently. After that, $|O_{\mathbf{S}^{(j)}}| = 1$ holds for all

**Algorithm 1:** One-Pass Training

**Data:** Clients $\mathcal{C}$, features $\mathbf{X}$, cluster size $K$, FedPC

**Result:** Trained fedPC

1 Set $\mathbf{M} = \mathbf{0}^{|\mathcal{C}| \times |\mathbf{X}|}$ and map = [];
2 $\mathbf{M}_{i,j} = 1$ if $X^{(j)}$ on client $i$;
3 **for** $j$, $\mathbf{u}$ *in enum. of distinct columns* $\mathcal{U}$ **do**
4      $\mathbf{S}^{(j)} = \{i : i \in \{1, \dots, |\mathbf{X}| \wedge \mathrm{all}(\mathbf{u} == \mathbf{M}_{:,i})\}\}$;
5      $O_{\mathbf{S}^{(j)}} = \mathrm{argwhere}(\mathbf{u} == 1)$;
6      map.append($\mathbf{S}^{(j)}$, $O_{\mathbf{S}^{(j)}}$);
7 sums = [];
8 **for** $\mathbf{S}^{(j)}$, $O_{\mathbf{S}^{(j)}}$ *in map* **do**
9      **if** $|O_{\mathbf{S}^{(j)}}| > 1$ **then**
10          s = fedPC.add_sum($\mathbf{S}^{(j)}$, $O_{\mathbf{S}^{(j)}}$);
11          sums.add(s)
12      **else**
13          client_clusters = cluster_local_data($O_{\mathbf{S}^{(j)}}$, $K$);
14 products = fedPC.add_products($P$);
15 **for** *prod in products* **do**
16      prod.children.add(sums);
17      **for** *client, clusters in client_clusters* **do**
18          prod.children.add_rand_subset(clusters);
19 fedPC.add_mixture_over_products(products);
20 fedPC.train_clients();
21 fedPC.infer_weights();
22 **return** fedPC

remaining $\mathbf{S}^{(j)}$. Also, the scope of the sums nodes introduced in the FedPC share no features with any of the remaining $\mathbf{S}^{(j)}$ since the server divided the feature space into disjoint subspaces. Therefore, we introduce $P$ product nodes to construct the remaining part of the FedPC. To this end, we divide the data of all subspaces $\mathbf{S}^{(j)}$ where $|O_{\mathbf{S}^{(j)}}| = 1$ holds into $K$ clusters **(Line 14)**. Each client learns a dedicated PC for each cluster. To ensure that the FedPC spans the entire feature space of the clients, the children of product nodes are set as follows: Each sum node introduced in the FedPC becomes a child of each product node. Additionally, for each $\mathbf{S}^{(j)}$ where $|O_{\mathbf{S}^{(j)}}| = 1$ holds, we randomly select a PC learned over one of the $K$ clusters s.t. the scope of each product node spans $\mathbf{X}$, and each PC representing a cluster is the child of at least one product node. Then, we build a mixture over all product nodes using a sum node **(Lines 15-20)**. Note that we seek to construct product nodes over independent clusters, which aligns with the maximum entropy principle (see App. D.1 for details). Once the FedPC is constructed, all client-sided PCs are learned. Since clients learn their PCs independently, each client can use an arbitrary learning algorithm (even different ones). As a last step, the network-sided parameters, i.e., the weights of network-sided sum nodes, of the FedPC are inferred **(Line 21-22)**. For each sum node $\mathsf{S}$, the weight $\mathbf{w}_{\mathsf{S}}^{(i)}$ associated with the $i$-th child (i.e., distribution) of $\mathsf{S}$ is set to $\frac{\rho(\mathsf{N}_i)}{\sum_i \rho(\mathsf{N}_i)}$. Here, $\rho(\mathsf{N}_i) = \sum_{\mathsf{C} \in \mathrm{ch}(\mathsf{N}_i)} |\mathbf{D}_{\mathsf{C}}|$ where $\mathbf{D}_{\mathsf{C}}$ is the dataset used to train the leaf $\mathsf{C}$. Note that this approach reduces horizontal FL to learning a mixture of the client's data distributions and vertical FL to learning a mixture over $P$ product nodes.

Due to the modularity of FCs, clients can learn parts of the final model independently, making FCs highly communication efficient. See App. C.1 for an analysis of the communication efficiency.

# 4 EXPERIMENTS

We now empirically evaluate FCs. We consider the scaling capabilities of FedPCs and the performance of FCs in horizontal, vertical, and hybrid FL. Therefore, we answer the following questions: **(Q1)** Do FedPCs effectively scale up PCs, thus yielding more expressive models? **(Q2)** How do FCs with different parameterizations perform on classification tasks compared to existing FL methods?

**Experimental Setup.** To see if FedPCs successfully scale up PCs, we follow Liu et al. (2024) and perform density estimation on three large-scale, high-resolution image datasets: Imagenet, Imagenet32 (both 1.2M samples), and CelebA (200K samples). The datasets were partitioned over 2-16 clients horizontally. We compare FedPCs to EiNets and Pyjuice. To evaluate FCs in FL scenarios, we selected three tabular datasets that cover various real-world application domains and data regimes. This includes one credit fraud dataset ($\sim 300K$ samples), a medical dataset (breast cancer detection; $< 1000$ samples), and the popular Income dataset ($> 1M$ samples). The selected datasets for FL cover low-data, medium-data, and large-data regimes (see App. F for more details). We compare FCs to FedAvg and SplitNN as widely used frameworks for horizontal and vertical FL, parameterized by the TabNet (Arik & Pfister, 2020) architecture tailored to tabular datasets. Additionally, we compare FCs to FedTree (Li et al., 2023b) since tree models excel at tabular datasets. For details, see App. F.

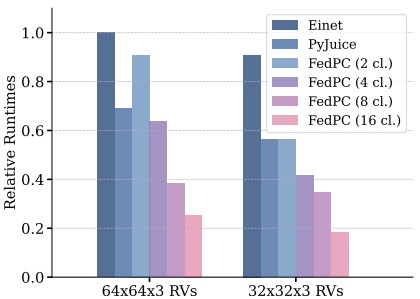

Figure 2: **FedPCs speed up training** due to parallel training on separate data partitions (here 64x64 and 32x32 images respectively).

|  | CelebA | Imagenet32 | Imagenet |
|---|---|---|---|
| EiNet | -3.42 ± 0.06 | -3.71 ± 0.04 | -3.73 ± 0.04 |
| PyJuice | -2.98 ± 0.02 | -3.60 ± 0.01 | -3.43 ± 0.02 |
| FedPC (2 cl.) | -2.87 ± 0.05 | -2.66 ± 0.02 | -3.12 ± 0.02 |
| FedPC (4 cl.) | -2.84 ± 0.05 | -2.56 ± 0.03 | -3.01 ± 0.03 |
| FedPC (8 cl.) | -2.76 ± 0.04 | -2.50 ± 0.03 | -2.97 ± 0.02 |
| FedPC (16 cl.) | **-2.68** ± 0.03 | **-2.45** ± 0.04 | **-2.90** ± 0.03 |

Table 1: **FedPCs outperform EiNets and PyJuice on density estimation tasks.** FedPCs achieve better results on density estimation tasks on three challenging image datasets (CelebA, Imagenet32, and Imagenet) because they can learn large models distributed across multiple machines. Results reported in nats (higher is better). Best in **bold**, 2nd best underlined.

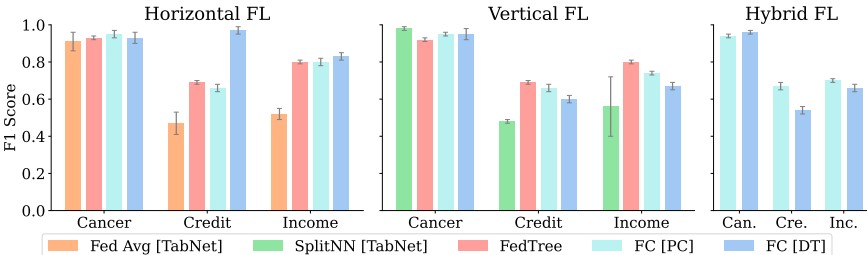

Figure 3: **FCs are competitive to prominent FL methods in all settings.** FCs achieve competitive performance on various classification tasks compared to prominent horizontal/vertical FL baselines. FCs also naturally handle hybrid FL without performance drops.

**(Q1) FedPCs effectively scale up PCs.** To examine whether FedPCs can be leveraged to scale up PCs effectively, we trained an EiNet, PyJuice, and FedPC on CelebA, Imagenet32, and Imagenet. All models used the Poon-Domingos (PD) architecture. FedPCs were parameterized with EiNets, and data was distributed among $\{2, 4, 8, 16\}$ clients. The FedPC client models and baseline models were selected to ensure that each fits within a single GPU (see App. F for system details). Einets and FedPCs were parameterized with Gaussian leave distributions, while PyJuice models were parameterized with Categorical distributions. The parameterizations were chosen based on empirical observations. For FedPC training, the images were randomly distributed horizontally, with each client holding approximately equally large subsets. Client models of FedPCs and all baselines were trained with EM. In Tab. 1, we show nats normalized over samples and dimensions achieved by EiNets, PyJuice, and FedPC on the test set. It can be seen that with an increasing number of participating clients and, thus, a larger model, the density estimation performance also increases on all three datasets. We posit that this is because larger models exhibit higher expressivity, allowing them to capture statistical characteristics of the data better than smaller models. Also, higher nats scores achieved on the test set by larger models indicate that no overfitting appeared due to more model parameters. However, note that more exhaustive scaling will likely lead to overfitting. Finding the optimal model size/number of clients in a principled way is beyond the scope of this work and is left for future endeavors. Besides better modeling performance, training time is reduced significantly with more clients (see relative runtimes in Fig. 2). FedPCs thus efficiently scale PCs to large datasets.

**(Q2) FCs achieve state of the art classification results in FL.** FCs can be parameterized with different models in the leaves. We examine two parameterizations to solve a federated classification task on three tabular datasets. First, we use the FedPC (FC [PC]), which can be used to solve discriminative tasks leveraging tractable computation of conditionals in PCs. The second FC parameterization we examine is decision trees (FC [DT]), representing an instantiation of a bagging model. To see how FCs perform in federated classification tasks, we compare FCs to well-known methods for horizontal FL and vertical FL. The experiments were conducted on tabular datasets covering various real-world application domains and distribution properties. We employ TabNet and FedTree as strong baselines. In the horizontal FL setting, TabNet was trained using FedAvg; in the vertical FL setting, it was trained in a SplitNN fashion (Ceballos et al., 2020). The results were compared against our one-pass training. FCs yield comparable or even better results than the selected baselines on all datasets (see Fig 3; App. G) while being significantly more flexible compared to the baselines.

## 5 CONCLUSION

In this work, we introduced federated circuits that hinge on an inherent connection between PCs and FL. We demonstrated that FCs naturally handle horizontal, vertical, and hybrid FL. Also, the training speed and expressivity of PCs can be increased by learning PCs on scale via FCs. Our framework allows for the seamless integration of various models on the client side, maintaining the relevance of FCs for FL and scaling of probabilistic models.

## ACKNOWLEDGEMENTS

This work is supported by the Hessian Ministry of Higher Education, Research, Science and the Arts (HMWK; projects "The Third Wave of AI"). Further, this work was supported from the National High-Performance Computing project for Computational Engineering Sciences (NHR4CES).

The Eindhoven University of Technology authors received support from their Department of Mathematics and Computer Science and the Eindhoven Artificial Intelligence Systems Institute.

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

# A  PRELIMINARIES

## A.1  PROBABILISTIC CIRCUITS

Peharz et al. (2015b) define a PC over random variables $\mathbf{X}$ as a tuple $(\mathcal{G}, \phi)$ where $\mathcal{G} = (V, E)$ is a rooted, Directed Acyclic Graph (DAG) and $\phi : V \to 2^{\mathbf{X}}$ is the *scope* function assigning a subset of random variables to each node in $\mathcal{G}$. For each internal node $\mathsf{N}$ of $\mathcal{G}$, the scope is defined as $\phi(\mathsf{N}) = \cup_{\mathsf{N}' \in \mathrm{ch}(\mathsf{N})} \phi(\mathsf{N}')$. Each leaf node $\mathsf{L}$ computes a distribution/density over its scope. All internal nodes of $\mathcal{G}$ are either a sum node $\mathsf{S}$ or a product node $\mathsf{P}$ where each sum node computes a convex combination of its children, i.e. $\mathsf{S} = \sum_{\mathsf{N} \in \mathrm{ch}(\mathsf{S})} w_{\mathsf{S},\mathsf{N}} \mathsf{N}$, and each product node computes a product of its children, i.e. $\mathsf{P} = \prod_{\mathsf{N} \in \mathrm{ch}(\mathsf{P})} \mathsf{N}$. To ensure tractability, a PC must be *decomposable*. Decomposability requires that for all $\mathsf{P} \in V$ it holds that $\phi(\mathsf{N}) \cap \phi(\mathsf{N}') = \emptyset$ where $\mathsf{N}, \mathsf{N}' \in \mathrm{ch}(\mathsf{P})$. To further ensure that a PC represents a valid distribution, *smoothness* must hold, i.e., for each sum $\mathsf{S} \in V$ it holds that $\phi(\mathsf{N}) = \phi(\mathsf{N}')$ where $\mathsf{N}, \mathsf{N}' \in \mathrm{ch}(\mathsf{S})$ (Poon & Domingos, 2011; Peharz et al., 2015a; Sánchez-Cauce et al., 2021).

# B  DISCUSSION ON ASSUMPTIONS

As a preliminary to FCs, we introduced two assumptions that allowed us to construct the FC framework. Here, we provide some more background on these assumptions. For clarity, let us state the assumptions again.

**Assumption 1** (Mixture Marginals). There exists a joint distribution $p$ such that the relation $\int_{\mathbf{X} \setminus \mathbf{X}_S} p(x) = \sum_{l \in L} q(L = l) \cdot p_S(x | L = l)$ holds for all $x \in \mathcal{X}$. Here, $\mathbf{X}_S \subseteq \mathbf{X}$ is a subset of the union of client random variables $\mathbf{X} = \cup_{c \in \mathcal{C}} \mathbf{X}_c$. Further, $\mathcal{X} = \times_{c \in \mathcal{C}} \mathcal{X}_c$ is the support of $\mathbf{X}$, each $p_S$ is defined over $\mathbf{X}_S \subseteq \mathbf{X}$ and $q$ is a prior over a latent $L$.

**Assumption 2** (Cluster Independence). Given disjoint sets of random variables $\mathbf{X}_1, \cdots, \mathbf{X}_n$ and a joint distribution $p(\mathbf{X}_1, \cdots, \mathbf{X}_n)$, assume that a latent $L$ can be introduced s.t. the joint can be represented as $p(\mathbf{X}_1, \cdots, \mathbf{X}_n) = \sum_l q(L = l) \prod_{i=1}^n p(\mathbf{X}_i | L = l)$ where $q$ is a prior distribution over the latent $L$.

Assumption 1 ensures that the data that resides on all participating clients is sufficient to learn $p(\mathbf{X})$, at least in the limit of infinite samples available. To illustrate, consider a subset of variables $\mathbf{X}_S \subseteq \mathbf{X}$ shared among all clients and its complement $\mathbf{X}_{S^-} = \mathbf{X} \setminus \mathbf{X}_S$. Assumption 1 ensures that the marginal $\int_{\mathbf{X}_{S^-}} p(\mathbf{X})$ is representable as a mixture of all client distributions $p_c(\mathbf{X}_S)$ over $\mathbf{X}_S$. If Assumption 1 would not hold, the information stored on the clients' data partitions would not be sufficient to learn $p(\mathbf{X})$.

There is also a PC perspective on this assumption. For this, let us introduce the induced tree representation of PCs from (Zhao et al., 2016):

**Definition 5.** *Induced Trees (Zhao et al., 2016). Given a complete and decomposable PC $s$ over $\mathbf{X} = \{X_1, \ldots, X_n\}$, $\mathcal{T} = (\mathcal{T}_V, \mathcal{T}_E)$ is called an induced tree PC from $s$ if*

1. *$\mathsf{N} \in \mathcal{T}_V$ where $\mathsf{N}$ is the root of $s$.*

2. *for all sum nodes $\mathsf{S} \in \mathcal{T}_V$, exactly one child of $\mathsf{S}$ in $s$ is in $\mathcal{T}_V$, and the corresponding edge is in $\mathcal{T}_E$.*

3. *for all product node $\mathsf{P} \in \mathcal{T}_V$, all children of $\mathsf{P}$ in $s$ are in $\mathcal{T}_V$, and the corresponding edges in $\mathcal{T}_E$.*

We can use Def. 5 to represent decomposable and complete PCs as mixtures (Zhao et al., 2016).

**Proposition 1** (Induced Tree Representation). *Let $\tau_s$ be the total number of induced trees in $s$. Then the output at the root of $s$ can be written as $\sum_{t=1}^{\tau_s} \prod_{(k,j) \in \mathcal{T}_{t_E}} w_{kj} \prod_{i=1}^n p_t(X_i = x_i)$, where $\mathcal{T}_t$ is the $t$-th unique induced tree of $s$ and $p_t(X_i)$ is a univariate distribution over $X_i$ in $\mathcal{T}_t$ as a leaf node.*

Using Prop. 1, we see that any decomposable and smooth PC can be represented as a mixture without any hierarchy, i.e., we can collapse the PC structure into a structure of depth one. Since marginalizing

over a decomposable and smooth PC yields another decomposable and smooth PC again, and since the marginalized PC can be represented as an induced tree, Assumption 1 is a standard assumption in the PC literature.

A key assumption in FL is that data cannot be exchanged among clients. However, dependencies among variables residing on different clients might still exist. Assumption 2 enables learning "hidden" dependencies between features partitioned over clients while keeping data private. Note that independence is only assumed within clusters in the data. Thus, the latent variable (which can be thought of as "cluster selectors") allows capturing dependencies among variables residing on different clients. Distributions of the form in Assumption 2 are strictly more expressive than the product distribution, thus allowing for more complex modeling.

Also, Assumption 2 can be viewed from a PC perspective. In popular structure learning algorithms such as LearnSPN Gens & Domingos (2013), a PC is learned by alternating data clustering with testing for independent subsets of features. Thus, the ultimate goal of algorithms like LearnSPN is to find clusters in which subsets of random variables are considered independent in order to maximize log-likelihood. Therefore, Assumption 2 is closely related to LearnSPN and, thus, a common assumption in PC modeling.

## C TRAINING ALGORITHM DETAILS

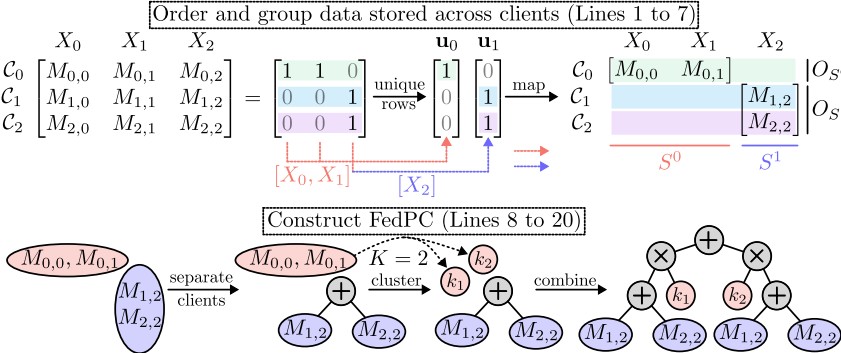

Figure 4: **One-Pass Training Visualized.** (Top) First, the matrix $\mathbf{M}$ is initialized, representing which features are held by which client. Feature subsets are constructed by considering distinct column vectors $\mathbf{u}$ of $\mathbf{M}$ that represent the same set of clients. This forms a mapping indicating which features are modeled as a mixture over clients. (Bottom) This mapping is utilized by forming mixtures over different clients sharing the same feature set via sum nodes. Features that are not shared over multiple clients will be clustered into $K$ clusters (here $K = 2$). The FedPC is formed by creating product nodes containing all sum nodes from the previous steps and at least one of the $K$ clusters. Lastly, the root node is inserted.

### C.1 ANALYSIS OF COMMUNICATION EFFICIENCY

As a key requirement for efficient training when learning models at scale on partitioned data, we now analyze the communication efficiency of FedPCs.

**Horizontal FL.** Assume a client set $\mathcal{C}$ where each client holds a model with $M$ parameters. Further, assume models are aggregated $K$ times during training ($K$ communication rounds). Then, model aggregation-based algorithms like FedAvg commonly used in horizontal FL send $\mathcal{O}(M \cdot |\mathcal{C}| \cdot K)$ messages over the network as each client sends $M$ model parameters to a server in each communication round. Training FedPCs with one-pass training, in contrast, only requires $\mathcal{O}(|\mathcal{C}| \cdot (M + 1))$ messages over the network as models are learned locally and independently, followed by setting the parameters ($\mathcal{O}(|\mathcal{C}|)$ messages) of the sum nodes and aggregating the model on the server ($\mathcal{O}(M|\mathcal{C}|)$ messages).

**Vertical FL.** In vertical settings, SplitNN-like architectures are commonly used. Assume training a SplitNN architecture for $E$ epochs that output a feature vector of size $F$ for each sample of a dataset with $S$ samples, vertically distributed over clients $\mathcal{C}$. The training requires sending $\mathcal{O}(E \cdot |\mathcal{C}| \cdot F \cdot S)$ messages over the network. In contrast, with one-pass training of FedPCs, each client learns a

dedicated PC with $M$ parameters for each of the $K$ clusters that are learned. The last layer of the FedPC is a mixture of $P$ products of clusters. The mixture parameters are set after training each client's model. Aggregating the learned models and setting the network-sided mixture parameters requires $\mathcal{O}(K \cdot M \cdot |\mathcal{C}| + P)$ messages to be sent. If $(K \cdot M + \frac{P}{|\mathcal{C}|}) < (E \cdot F \cdot S)$ holds, training FedPCs is more communication efficient than training SplitNN-like architectures. In practice, this is likely to hold: The number of clusters is usually smaller than 100 while feature vectors can have hundreds of dimensions (i.e., $F > 100$). Further, models should have fewer parameters than samples in the dataset to ensure generalization (i.e., $M < S$). $P$ can be set to an arbitrary value, depending on $|\mathcal{C}|$ and the data. App. E provides more details and an intuition on communication costs.

**Hybrid FL.** In hybrid FL, FedPCs are trained on several subspaces: Some exist on all or a subset of clients (denoted as $R_s$) and some are only available on one client (denoted as $R_d$). Further denote communication costs of FedPCs in horizontal FL and vertical FL as $C_h$ and $C_v$, respectively. Since the training procedure in hybrid cases essentially performs horizontal FL on shared feature spaces and vertical FL on disjoint feature spaces, $\mathcal{O}(|R_s| \cdot C_h + |R_v| \cdot C_v)$ messages are sent over the network during training.

# D    PROOFS

In this section we give full proofs for our propositions in the paper.

## D.1    FEDPCS AND PRINCIPLE OF MAXIMUM ENTROPY

Assumption 2 aligns with the principle of maximum entropy: we aim to find the joint distribution with maximum entropy *within* clusters while allowing for dependencies among clients' random variables and ensuring the marginals for each client are preserved. Although multiple joint distributions can preserve the marginals, non-maximal entropy solutions introduce additional assumptions or prior knowledge, limiting flexibility. By assuming independence of all variables within a cluster, we efficiently construct the maximum entropy distribution via a mixture of product distributions. For independent variables, the product distribution maximizes entropy, as can be shown by leveraging the joint and conditional differential entropy. Given random variables $\mathbf{X} = X_1, \ldots, X_n$ and a density $p$ defined over support $\mathcal{X} = \mathcal{X}_1 \times \cdots \times \mathcal{X}_n$, the joint differential entropy is defined as:

$$h(\mathbf{X}) = \int_{\mathcal{X}} p(x_1, \ldots, x_n) \log p(x_1, \ldots, x_n) \tag{1}$$

The conditional differential entropy for two sets of random variables $\mathbf{X}$ and $\mathbf{Y}$ and a joint distribution $p(\mathbf{X}, \mathbf{Y})$ defined over support $\mathcal{X} \times \mathcal{Y}$ is defined analogously:

$$h(\mathbf{X}|\mathbf{Y}) = \int_{\mathcal{X}, \mathcal{Y}} p(\mathbf{x}, \mathbf{y}) \log p(\mathbf{x}|\mathbf{y}) \tag{2}$$

Given two sets of random variables $\mathbf{X}, \mathbf{Y}$ with densities $p(\mathbf{X})$ and $p(\mathbf{Y})$ and support $\mathcal{X}, \mathcal{Y}$ respectively, the joint $p(\mathbf{X}, \mathbf{Y}) = p(\mathbf{X}) \cdot p(\mathbf{Y})$ is the maximum entropy distribution if $\mathbf{X}$ and $\mathbf{Y}$ are mutually independent.

*Proof.* We consider the two cases that $\mathbf{X}$ and $\mathbf{Y}$ are mutually independent and that they are not mutually independent. The joint entropy can be written as $h(\mathbf{X}, \mathbf{Y}) = h(\mathbf{X}|\mathbf{Y}) + h(\mathbf{Y})$. In the case of mutual independence, this reduces to $h(\mathbf{X}, \mathbf{Y}) = h(\mathbf{X}) + h(\mathbf{Y})$. Hence it has to be shown that $h(\mathbf{X}|\mathbf{Y}) < h(\mathbf{X})$ holds if $\mathbf{X}$ and $\mathbf{Y}$ are not mutually independent:

$$h(\mathbf{X}|\mathbf{Y}) < h(\mathbf{X})$$

$$\equiv -\int_{\mathcal{X}, \mathcal{Y}} p(\mathbf{x}, \mathbf{y}) \log p(\mathbf{x}|\mathbf{y}) < -\int_{\mathcal{X}, \mathcal{Y}} p(\mathbf{x}, \mathbf{y}) \log p(\mathbf{x})$$

$$\equiv -\left( \int_{\mathcal{X}, \mathcal{Y}} p(\mathbf{x}, \mathbf{y}) \log p(\mathbf{x}|\mathbf{y}) - \int_{\mathcal{X}, \mathcal{Y}} p(\mathbf{x}, \mathbf{y}) \log p(\mathbf{x}) \right) < 0$$

$$\equiv -\left( \int_{\mathcal{X}, \mathcal{Y}} p(\mathbf{x}, \mathbf{y}) \log \frac{p(\mathbf{x}|\mathbf{y})}{p(\mathbf{x})} \right) < 0$$

Since $\mathbf{X} \perp\!\!\!\perp \mathbf{Y}$ holds where $\perp\!\!\!\perp$ means mutual independence, $\frac{p(\mathbf{x}|\mathbf{y})}{p(\mathbf{x})} \neq 1$ at least for some $\mathbf{x}, \mathbf{y}$. Since the mutual independence $I(\mathbf{X}, \mathbf{Y}) = \int_{\mathcal{X}, \mathcal{Y}} p(\mathbf{x}, \mathbf{y}) \log \frac{p(\mathbf{x}, \mathbf{y})}{p(\mathbf{x}) \cdot p(\mathbf{y})}$ can be represented as $I(\mathbf{X}, \mathbf{Y}) = h(\mathbf{X}) - h(\mathbf{X}|\mathbf{Y})$, $I(\mathbf{X}, \mathbf{Y}) \geq 0$ holds and $-\left( \int_{\mathcal{X}, \mathcal{Y}} p(\mathbf{x}, \mathbf{y}) \log \frac{p(\mathbf{x}|\mathbf{y})}{p(\mathbf{x})} \right) = h(\mathbf{X}|\mathbf{Y}) - h(\mathbf{X})$ it follows that $h(\mathbf{X}) > h(\mathbf{X}|\mathbf{Y})$.

$\square$

# E  COMMUNICATION EFFICIENCY

Communication efficiency is a critical property when it comes to learning models across multiple machines, as it is done in FL. Here, in addition to our theoretical results, we more intuitively provide further details on the communication efficiency of FCs. For that, we plot the communication cost in Megabytes (MB) required to train a FedPC vs. FedAvg/SplitNN in horizontal/vertical FL settings with datasets of different sizes (1M and 100M samples). Regardless of the number of samples in the dataset, FedPCs are more communication efficient compared to our baselines in both horizontal and vertical settings (see Fig. 5).

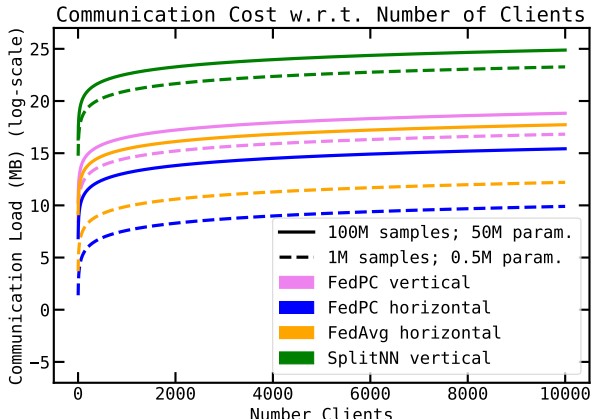

Figure 5: **FedPCs are communication-efficient.** We compare communication cost in Megabytes (MB) sent over the network during one full training of a model (0.5M/50M parameters) on a dataset (1M/100M samples) using results from Section 3.4. Results are shown on log-scale. It can be seen that FedPCs significantly reduce communication cost of training.

# F  EXPERIMENTAL DETAILS

**Experimental Setup.** To see if FedPCs, an instantiation of FCs, successfully scale up PCs, we follow Liu et al. (2024) and perform density estimation on three large-scale, high-resolution image datasets: Imagenet, Imagenet32 (both 1.2M samples), and CelebA (200K samples). The datasets were partitioned over 2-16 clients horizontally. We compare FedPCs to EiNets and Pyjuice.

To evaluate FCs in FL scenarios, we selected three tabular datasets that cover various application domains and data regimes present in the real world: one credit fraud dataset ($\sim$ 300K samples), a medical dataset (breast cancer detection; $<$ 1000 samples), and the popular Income dataset ($>$ 1M samples). The selected datasets for FL cover low-data, medium-data, and large-data regimes (see App. F for more details). Both balanced (breast cancer) and imbalanced (income, credit) datasets are included in our evaluation. We selected tabular datasets as they are well suited to investigate FCs in horizontal, vertical, and hybrid settings and represent various real-world applications. We compare FCs to multiple strong and widely used baselines. As a neural network architecture parameterization, we use TabNet (Arik & Pfister, 2020) which is tailored to tabular datasets. We train the networks with the widely used FedAvg (horizontal FL) and SplitNN (vertical FL) frameworks. Additionally, we compare FCs to FedTree (Li et al., 2023b) since tree models excel at tabular datasets.

## F.1  DATASETS

The following describes the datasets used in our experiments. If not stated differently, the datasets were distributed across clients as follows:

In horizontal cases, we either split samples randomly across clients (done for all binary classification tasks) or we distribute a subset of the dataset corresponding to a certain label (e.g. the 0 in MNIST) to one client.

In vertical cases, we split tabular datasets randomly along the feature-dimension, i.e. each client gets all samples but a random subset of features assigned. For image data, we split the images into non-overlapping patches which were then distributed to the clients.

In hybrid cases, we split tabular datasets along both, the feature and the sample-dimension. We do this s.t. at least two clients have at least one randomly chosen feature in commeon (but hold different samples thereof). For image data, we split images into overlapping patches, sample a subset of the dataset and assign the resulting subsets to clients.

**Income Dataset.** We used the Income dataset from `https://www.kaggle.com/datasets/wenruliu/adult-income-dataset`. This dataset represents a binary classification problem with 14 features and approximate 450K samples in the train and 900 samples in the test set. We encoded discrete variables to numerical values using TargetEncoder from sklearn. Additionally, missing values were imputed using the median of the corresponding feature. Further we standardized all features.

**Breast Cancer Dataset.** We used the Breast Cancer dataset from `https://www.kaggle.com/datasets/uciml/breast-cancer-wisconsin-data`. It represents a binary classification problem with 31 features and 570 samples. We split the dataset into 450 training samples and 120 test samples. We standardized all features for training.

**Credit Dataset.** We used the Give Me Some Credit dataset from `https://www.kaggle.com/c/GiveMeSomeCredit`. The dataset represents a binary classification task with 10 features, 1.5M training samples and 100K test samples. We encoded discrete variables to numerical values using TargetEncoder from sklearn. Additionally, missing values were imputed using the median of the corresponding feature. Further we standardized all features.

**MNIST.** We used the MNIST dataset provided by pytorch. It contains 70K hand-written digits between 0 and 9 as 28x28 images (60K train, 10K test). We standardized all features as preprocessing.

**Imagenet/Imagenet32.** We used the Imagenet dataset provided by pytorch. It consists of about 1.2M images showing objects of 1000 classes. The images come in different resolutions; we resized each image to 64x64 (Imagenet) and 32x32 (Imagenet32) pixels, applied center cropping, and standardized all features as preprocessing. We distributed all images randomly across clients.

### F.2 DISCRETIZATION

In our experimental setup, FCs and Einets were parameterized with Gaussian leaves and fitted on RGB image data. Since image data is discrete (takes integer values from 0-255) and Gaussians are defined over a continuous domain and thus define a probability *density* rather than a probability *mass* function, we have to discretize the Gaussian leaves to obtain the probability for a given image $\mathbf{x}$. Therefore, we construct 255 buckets, discretizing a Gaussian with parameters $\mu$ and $\sigma$ by computing the probability mass as $p(x) = \Phi(\frac{x-\mu+\frac{1}{255}}{\sigma}) - \Phi(\frac{x-\mu}{\sigma})$. Since the probabilistic semantics of PCs holds for densities and probability mass functions, the computation graph will remain fixed.

### F.3 HYPERPARAMETERS

The following tables show the setting of all relevant hyperparameters for each dataset and FL setting.

| FL-Setting | Dataset | Structure | Threshold | min_num_instances | glueing |
|---|---|---|---|---|---|
| horizontal | Income | learned | 0.3 | 200 | - |
| | Credit | learned | 0.5 | 200 | - |
| | Cancer | learned | 0.4 | 300 | - |
| vertical | Income | learned | 0.4 | 100 | combinatorial |
| | Credit | learned | 0.5 | 50 | combinatorial |
| | Cancer | learned | 0.4 | 300 | combinatorial |
| hybrid | Income | learned | 0.4 | 100 | combinatorial |
| | Credit | learned | 0.5 | 50 | combinatorial |
| | Cancer | learned | 0.4 | 300 | combinatorial |

Table 2: Hyperparameters used in our experiments for all tabular datasets.

|  | MNIST | Imagenet(32) | CelebA |
|---|---|---|---|
| num_epochs | 5 | 25 | 10 |
| batch_size | 64 | 64 | 64 |
| online_em_frequency | 5 | 10 | 10 |
| online_em_stepsize | 0.1 | 0.25 | 0.25 |
| Structure | poon-domingos | poon-domingos | poon-domingos |
| pd_num_pieces | 4 | 4 | 4 |
| K | 10 | 120 | 120 |
| Leaf Distribution | Gaussian | Gaussian | Gaussian |
| min_var | $1 \cdot 10^{-3}$ | $1 \cdot 10^{-3}$ | $1 \cdot 10^{-3}$ |
| max_var | $1 \cdot 10^{-7}$ | $1 \cdot 10^{-7}$ | $1 \cdot 10^{-7}$ |

Table 3: Hyperparameters used in our experiments for image datasets.

## F.4 HARDWARE

All experiments were conducted on Nvidia DGX machines with Nvidia A100 (40GB) GPUs, AMD EPYC 7742 64-Core Processor and 2TiB of RAM.

## G FURTHER RESULTS

Here, we provide further experimental details on FCs.

**FedPCs learn joint distributions over partitioned data in less time.** First, we validate that FedPCs correctly and efficiently perform density estimation on partitioned datasets distributed over multiple clients. To this end, multiple datasets were distributed over a set of clients corresponding to horizontal (5 clients), vertical (2 clients), and hybrid FL (2 clients). To demonstrate that FedPCs are also robust against label shifts, a common regime in FL, each client received data from only a subset of classes in the horizontal case, and local PCs were learned over the client samples. In the vertical case, we split data s.t. feature spaces of clients are disjoint, but each client holds the same samples. In hybrid settings, data was distributed s.t. both feature- and sample-spaces among clients have overlaps (but no full overlap). For all tabular datasets, the leaves of the FedPC were parameterized with MSPNs (Molina et al., 2018), a member of the PC model family capable of performing density estimation on mixed data domains (i.e., continuous and discrete random variables). We chose MSPNs as the centralized models, which were learned using LEARNSPN, a recursive greedy structure learning algorithm for SPNs Gens & Domingos (2013). For MNIST, EiNets with Gaussian densities were used as PC instantiations in all settings. Note that FedPCs were chosen to approximately match the size of centralized models, i.e., no model upscaling was performed.

Tab. 4 compares log-likelihood scores and relative runtime of centralized training of a PC on the full datasets with log-likelihood scores and relative runtimes achieved by FedPC in different FL settings. FedPCs successfully reproduce the results of centralized PCs on tabular datasets while being tremendously faster in training. This validates our approach and we answer **(Q1)** affirmatively.

|  | Log-Likelihood | | | | Relative Runtime | | | |
|---|---|---|---|---|---|---|---|---|
|  | cent | horizontal | vertical | hybrid | cent | horizontal | vertical | hybrid |
| MNIST | $3352_{\pm 3.5}$ | $3350_{\pm 3.2}$ | $3351_{\pm 3.8}$ | $3349_{\pm 3.7}$ | 1.0 | $\mathbf{0.07}_{\pm \mathbf{0.01}}$ | $0.13_{\pm 0.01}$ | $0.13_{\pm 0.02}$ |
| Income | $-11.5_{\pm 0.1}$ | $-11.4_{\pm 3.5}$ | $-11.9_{\pm 3.3}$ | $-12.0_{\pm 1.5}$ | 1.0 | $\mathbf{0.17}_{\pm \mathbf{0.02}}$ | $0.236_{\pm 0.01}$ | $0.21_{\pm 0.02}$ |
| Cancer | $-38.9_{\pm 0.3}$ | $-38.5_{\pm 1.1}$ | $-38.6_{\pm 0.5}$ | $-38.7_{\pm 1.5}$ | 1.0 | $\mathbf{0.21}_{\pm \mathbf{0.07}}$ | $0.35_{\pm 0.05}$ | $0.35_{\pm 0.1}$ |
| Credit | $-12.8_{\pm 1.0}$ | $-13.1_{\pm 0.5}$ | $-12.5_{\pm 2.3}$ | $-12.5_{\pm 1.3}$ | 1.0 | $0.42_{\pm 0.05}$ | $\mathbf{0.31}_{\pm \mathbf{0.09}}$ | $0.40_{\pm 0.13}$ |

Table 4: **FedPCs speed up training while retaining model performance.** We trained PCs in a centralized setting (cent.) and in all FL settings (using FedPCs) on different datasets and the same structure learning algorithm. We find that FedPCs tremendously speed up training while there is no reduction in log-likelihood. This demonstrates that PCs can be learned in federated settings (for MNIST, log densities are reported). We report relative runtime where centralized runtime is 1.0.

**One-pass training retains performance.** To see how the proposed one-pass training compares to training PCs with standard optimization algorithms such as EM, we define an FL setup where

data exchange is allowed. This is necessary as we have to train the PC and FedPC architecture with EM to compare to our one-pass procedure. We used RAT-SPNs (Peharz et al., 2020b) as leaf parameterizations of the FedPC. Then, we trained a FedPC using standard EM (i.e., data exchange was allowed) and another FedPC with the same FedPC architecture on a vertically split dataset using our one-pass procedure. We report the final average log-likelihood of the test dataset, both for EM training and one-pass training (see Tab. 5). It can be seen that there is no significant decrease in log-likelihood in any case. Interestingly, the one-pass training seems even to be better than EM. We suspect that that it is easier to solve the subtasks of local training independently instead of jointly optimizing all parameters of the FedPC. Hence, our results indicate that one-pass training is preferable since it is communication efficient. one-pass training can be used instead of the more costly EM scheme.

|  | EM | one-pass |
|---|---|---|
| Synth. Data | $-53.6 \pm 1.3$ | $-53.2 \pm 1.2$ |
| Income | $-18.5 \pm 0.1$ | $-18.0 \pm 0.5$ |
| Breast-Cancer | $-52.3 \pm 0.2$ | $-55.7 \pm 0.2$ |
| Credit | $-26.7 \pm 1.2$ | $-28.3 \pm 0.4$ |

Table 5: **One-pass training retains performance.** We trained the same FedPC architecture on various datasets using EM and one-pass training in a vertical setting. The average log-likelihood value of the hold-out test set across 10 runs is reported.

**FL Classification Results.** We compare FCs to several baselines in horizontal, vertical, and hybrid FL. In horizontal FL, we compare against FedAvg (using TabNet (Arik & Pfister, 2020)) and FedTree (Li et al., 2023b); in vertical FL, we compare against SplitNN (also using TabNet) and FedTree. In hybrid FL, we compare different parameterizations of FCs (FedPCs and FCs parameterized with decision trees). We find that FCs are competitive or outperforming the selected baselines in all FL settings (see Tab. 6). This makes them a very flexible FL framework that still yields high-performing models.

|  |  | Cancer | | Credit | | Income | |
|---|---|---|---|---|---|---|---|
|  |  | Acc. | F1 | Acc. | F1 | Acc. | F1 |
| Horizontal FL | FedAvg [TabNet] (5 cl.) | $0.92 \pm 0.03$ | $0.92 \pm 0.03$ | $0.71 \pm 0.11$ | $0.48 \pm 0.04$ | $0.68 \pm 0.06$ | $0.51 \pm 0.03$ |
|  | FedAvg [TabNet] (10 cl.) | $0.92 \pm 0.04$ | $0.91 \pm 0.05$ | $0.56 \pm 0.12$ | $0.47 \pm 0.06$ | $0.64 \pm 0.06$ | $0.52 \pm 0.03$ |
|  | FedTree (5 cl.) | $0.93 \pm 0.01$ | $0.92 \pm 0.01$ | $0.91 \pm 0.01$ | $0.63 \pm 0.01$ | $0.88 \pm 0.01$ | $0.82 \pm 0.02$ |
|  | FedTree (10 cl.) | $0.94 \pm 0.01$ | $0.93 \pm 0.01$ | $0.92 \pm 0.01$ | $0.69 \pm 0.01$ | $0.87 \pm 0.01$ | $0.80 \pm 0.01$ |
|  | FC [PC] (5 cl.) | $0.98 \pm 0.01$ | $0.98 \pm 0.01$ | $0.93 \pm 0.02$ | $0.68 \pm 0.02$ | $0.87 \pm 0.02$ | $0.80 \pm 0.01$ |
|  | FC [PC] (10 cl.) | $0.95 \pm 0.02$ | $0.95 \pm 0.02$ | $0.93 \pm 0.01$ | $0.66 \pm 0.02$ | $0.87 \pm 0.01$ | $0.80 \pm 0.02$ |
|  | FC [DT] (5 cl.) | $0.95 \pm 0.03$ | $0.93 \pm 0.02$ | $0.92 \pm 0.01$ | $0.67 \pm 0.01$ | $0.89 \pm 0.01$ | $0.83 \pm 0.01$ |
|  | FC [DT] (10 cl.) | $0.95 \pm 0.02$ | $0.93 \pm 0.03$ | $0.92 \pm 0.01$ | $0.97 \pm 0.02$ | $0.89 \pm 0.01$ | $0.83 \pm 0.02$ |
|  | SplitNN [TabNet] | - | - | - | - | - | - |
| Vertical FL | SplitNN [TabNet] (2 cl.) | $0.98 \pm 0.01$ | $0.98 \pm 0.01$ | $0.93 \pm 0.01$ | $0.48 \pm 0.01$ | $0.56 \pm 0.25$ | $0.42 \pm 0.17$ |
|  | SplitNN [TabNet] (3 cl.) | $0.98 \pm 0.01$ | $0.98 \pm 0.01$ | $0.93 \pm 0.01$ | $0.48 \pm 0.01$ | $0.62 \pm 0.20$ | $0.56 \pm 0.16$ |
|  | FedTree (2 cl.) | $0.94 \pm 0.01$ | $0.93 \pm 0.01$ | $0.92 \pm 0.01$ | $0.69 \pm 0.02$ | $0.87 \pm 0.01$ | $0.80 \pm 0.01$ |
|  | FedTree (3 cl.) | $0.93 \pm 0.01$ | $0.92 \pm 0.01$ | $0.92 \pm 0.01$ | $0.69 \pm 0.01$ | $0.87 \pm 0.01$ | $0.80 \pm 0.01$ |
|  | FC [PC] (2 cl.) | $0.96 \pm 0.01$ | $0.96 \pm 0.01$ | $0.92 \pm 0.01$ | $0.67 \pm 0.01$ | $0.84 \pm 0.02$ | $0.74 \pm 0.01$ |
|  | FC [PC] (3 cl.) | $0.95 \pm 0.01$ | $0.95 \pm 0.01$ | $0.92 \pm 0.01$ | $0.66 \pm 0.02$ | $0.84 \pm 0.01$ | $0.74 \pm 0.01$ |
|  | FC [DT] (2 cl.) | $0.96 \pm 0.01$ | $0.96 \pm 0.02$ | $0.93 \pm 0.01$ | $0.60 \pm 0.02$ | $0.83 \pm 0.02$ | $0.67 \pm 0.02$ |
|  | FC [DT] (3 cl.) | $0.95 \pm 0.01$ | $0.95 \pm 0.03$ | $0.93 \pm 0.01$ | $0.60 \pm 0.02$ | $0.82 \pm 0.02$ | $0.67 \pm 0.02$ |
|  | FedAvg [TabNet] | - | - | - | - | - | - |
| Hybrid FL | FC [PC] (2 cl.) | $0.94 \pm 0.01$ | $0.94 \pm 0.01$ | $0.92 \pm 0.01$ | $0.67 \pm 0.01$ | $0.82 \pm 0.02$ | $0.71 \pm 0.01$ |
|  | FC [PC] (3 cl.) | $0.94 \pm 0.01$ | $0.94 \pm 0.01$ | $0.92 \pm 0.01$ | $0.67 \pm 0.02$ | $0.80 \pm 0.01$ | $0.70 \pm 0.01$ |
|  | FC [DT] (2 cl.) | $0.96 \pm 0.01$ | $0.96 \pm 0.02$ | $0.93 \pm 0.01$ | $0.60 \pm 0.02$ | $0.82 \pm 0.02$ | $0.66 \pm 0.02$ |
|  | FC [DT] (3 cl.) | $0.96 \pm 0.01$ | $0.96 \pm 0.01$ | $0.93 \pm 0.01$ | $0.54 \pm 0.02$ | $0.82 \pm 0.02$ | $0.66 \pm 0.02$ |
|  | FedAvg [TabNet] | - | - | - | - | - | - |
|  | SplitNN [TabNet] | - | - | - | - | - | - |
|  | FedTree | - | - | - | - | - | - |

Table 6: **All Classification results of FL experiments.** Here, we show the detailed performances of FC, FedAvg, and SplitNN in all three FL settings. It can be seen that FCs, while being much more flexible than our baselines, still achieve competitive or better results on various classification tasks.

