# OpenReview forum: "Federated Circuits: A Unified Framework for Scalable and Efficient Federated Learning"
_ICLR.cc/2025/Workshop/MCDC — MCDC @ ICLR 2025_

### Official Review · Reviewer_A9Bw · 2025-02-27

**Rating:** 6
**Confidence:** 3
**Fit:** 4

**Summary:**

This paper introduces Federated Circuits (FCs) and Federated Probabilistic Circuits (FedPCs) as a novel FL framework flexible to diverse kind of FL scenarios (horizontal, vertal, hybrid) which promises increased communication efficiency and parallelization in FL.
By showing the similarities between data distribution in various FL settings and concepts in PCs, authors devise a training procedure to learn FCs via a FL procedure.

**Reason For Giving A Higher Score:**

Interesting approach, very different from common ones

**Reason For Giving A Lower Score:**

Some crucial parts are unclear, and this partially impedes reviewing the work.

**Strengths And Weaknesses:**

Strenghts:
- The formalization appears sound, and the parallelism with PCs is mostly clear. The approach seems novel and interesting to me.
- The experiments seem convincing in showing the potential of this novel approach
- The paper is carefully written and the code is already open source. The discussion in the supplementary is useful and well presented

Weaknesses:
- While the effort in writing the manuscript is clear, the proposed is not very clear, in particular for readers who don't know about probabilistic circuits. As a consequence, while the general high level idea is conveyed, the detail of what happens during client training and what it is sent over the network are not clear. In particular, while authors claim that the proposed method is compliant with privacy requirements of FL, this matter is not explained.
- Relationship with classical approaches is not discussed: this agains insists on clarity, because it is not clear if, in a given scenarios (let's say horizontal FL) this approach has advatages related to model quality.
- It is not clear which kind of problems this approach can work with: authors presented classification and density estimation, but the paper would benefit a discussion of pros/cons of the proposed approach

**Suggestions:**

Suggestions:
- Invest more in explanations regarding pros/cons of the algorithms, its applicability and relationship with predominant approaches
- Explain in detail what computations clients do, what they exchange and why it is still privacy preserving
- Please explain more in detail the practical meaning of the assumptions used and how they relate to standard assumptions
- Evaluate to provide a proof about the effect of data heterogeneity: if my understanding is correct, it should be possible to show that FCs are not affected by data heterogeneity, offering a substantial advantage over standard algorithms that require advanced techniques to handle heterogeneity

---

### Official Review · Reviewer_2rae · 2025-02-27

**Rating:** 7
**Confidence:** 4
**Fit:** 5

**Summary:**

This paper introduces Federated Circuits (FCs), a unified framework for federated learning that leverages the semantics of probabilistic circuits (PCs) to jointly address horizontal, vertical, and hybrid federated learning (FL) settings. By re-framing FL as a density estimation task, the authors propose a novel approach that builds modular, communication‐efficient models—termed FedPCs—through the use of sum nodes (for aggregating client-specific distributions) and product nodes (for integrating disjoint feature spaces). A key contribution is the design of a one‐pass training algorithm that significantly reduces communication overhead while scaling up the expressivity of PCs across distributed datasets. The paper supports its claims with extensive experiments on both large-scale image datasets (e.g., Imagenet, CelebA) and tabular datasets (e.g., credit, medical, income), comparing against strong baselines such as EiNets, PyJuice, FedAvg, SplitNN, and FedTree.

**Reason For Giving A Higher Score:**

The paper makes a significant conceptual contribution by unifying disparate FL settings under a single framework and provides strong empirical evidence to support its claims. The approach is innovative, and the communication efficiency improvements, coupled with solid experimental results, make it a promising direction for scalable FL.

**Reason For Giving A Lower Score:**

The framework, though innovative, is limited by its evaluation on a small number of clients and lacks thorough analysis of performance under extreme non-IID conditions.

**Strengths And Weaknesses:**

**Strengths:**

- Novel Concept: The paper presents an innovative idea by linking the semantics of probabilistic circuits to federated learning, thereby offering a unified framework for multiple FL settings.
- Unified Approach: FCs elegantly handle horizontal, vertical, and hybrid FL within a single framework, which could simplify and generalize current FL methodologies.
- Communication Efficiency: The one-pass training algorithm significantly reduces communication overhead—a key advantage in federated scenarios.
- Extensive Empirical Evaluation: Experiments on both image and tabular data are thorough, demonstrating scalability and performance gains.
- Theoretical Grounding: The paper provides a solid theoretical basis by leveraging properties of PCs and by analyzing communication costs.

**Weaknesses:**

- Assumptions: The approach relies on modeling assumptions (e.g., mixture marginals and cluster independence) that could benefit from further discussion regarding their practical validity.
- Comparative Analysis: Broader comparisons with more recent state-of-the-art FL methods could help position the contribution more clearly within the literature.
- Ablation Studies: While extensive experiments are presented, additional ablation studies (e.g., on the effect of the number of clients, sensitivity to hyperparameters) would strengthen the evaluation.
- Scalability to Heterogeneous Data: It remains to be seen how robust the method is when faced with highly heterogeneous client data distributions or when scaling to an even larger number of clients.

**Suggestions:**

- Scalability to Massive Client Numbers: Evaluate the framework in settings with thousands of clients to assess communication overhead, model aggregation challenges, and robustness to client dropouts or asynchronous updates.
- Handling Extreme Non-IID Data: Extend experiments and analysis to include more severe non-IID scenarios. Consider introducing simulated heterogeneity or using real-world federated datasets to examine convergence, stability, and performance degradation.
- Clarification of Assumptions: Provide a more detailed discussion of the underlying assumptions (mixture marginals and cluster independence), including potential limitations when these assumptions are violated

---

### Official Review · Reviewer_Jyoh · 2025-03-03

**Rating:** 7
**Confidence:** 2
**Fit:** 5

**Summary:**

The submission titled "Federated Circuits: A Unified Framework for Scalable and Efficient Federated Learning" introduces Federated Circuits (FCs), a framework that unifies horizontal, vertical, and hybrid federated learning (FL) by treating FL as a density estimation problem over distributed datasets. The authors claim that FCs enable scalable and communication-efficient learning. The submission includes extensive experimental results showing FCs' performance on multiple tasks compared to existing methods. It also proposes a one-pass training scheme for Federated Probabilistic Circuits (FedPCs).

**Reason For Giving A Higher Score:**

The paper presents an innovative and unified framework for federated learning. It also includes a thorough empirical validation.

**Reason For Giving A Lower Score:**

N/A

**Strengths And Weaknesses:**

Strengths:
- Unified Framework: The paper introduces a novel framework that unifies horizontal, vertical, and hybrid federated learning (FL), which to the best of my knowledge is a significant advancement in the field.
- Empirical Validation: The paper provides thorough experimental results that show FCs outperform existing methods on large-scale density estimation tasks and achieve competitive results on classification tasks.
- Public Availability: The authors have made the code publicly available, promoting transparency and reproducibility in research.

Weakness:
- Complexity: The proposed framework and training scheme are complex, which might pose challenges for implementation and understanding by practitioners who are not familiar with probabilistic circuits or federated learning.

**Suggestions:**

Although the paper does not have a dedicated limitations section, it does implicitly discuss limitations in other sections. It would be beneficial to make this discussion more explicit. Similarly, future work is briefly mentioned, but it could be more directly discussed.

The language used is clear and correctly spelled throughout the document, but some phrases could be improved for readability. Software like Grammarly could help identify and reformulate such phrases.

---

### Decision · Program_Chairs · 2025-03-06

**Decision:**

Accept

**Comment:**

This work proposes to combine probabilitic circuits and federated learning, and modelize a distributed optimization as a density estimation over distributed dataset, a relevant topic to this workshop. All reviewers recommend acceptance and we're pleased to accept this work to the workshop.